# The SARS-CoV-2 Spike Protein Receptor-Binding Domain Expressed in Rice Callus Features a Homogeneous Mix of Complex-Type Glycans

**DOI:** 10.3390/ijms25084466

**Published:** 2024-04-18

**Authors:** Guillermo Sobrino-Mengual, Victoria Armario-Nájera, Juliette Balieu, Marie-Laure Walet-Balieu, Andrea Saba-Mayoral, Ana M. Pelacho, Teresa Capell, Paul Christou, Muriel Bardor, Patrice Lerouge

**Affiliations:** 1Department of Agricultural and Forest Sciences and Engineering, University of Lleida, Agrotecnio CERCA Center, 25003 Lleida, Spain; guillermo.sobrino@udl.cat (G.S.-M.); victoria.armario@udl.cat (V.A.-N.); andrea.saba@udl.cat (A.S.-M.); anamaria.pelacho@udl.cat (A.M.P.); teresa.capell@udl.cat (T.C.); paul.christou@udl.cat (P.C.); 2GlycoMEV UR 4358, SFR Normandie Végétal FED 4277, Innovation Chimie Carnot, IRIB, GDR CNRS Chemobiologie, RMT BESTIM, Université de Rouen Normandie, F-76000 Rouen, France; juliette.balieu@univ-rouen.fr; 3INSERM, CNRS, HeRacLeS US51 UAR2026, PISSARO, Université de Rouen Normandie, F-76000 Rouen, France; marie-laure.walet-balieu@univ-rouen.fr (M.-L.W.-B.); muriel.bardor@univ-rouen.fr (M.B.); 4Catalan Institute for Research and Advanced Studies (ICREA), 08010 Barcelona, Spain

**Keywords:** plant molecular farming, SARS-CoV-2, receptor-binding domain, glycan profile, biologics, spike protein

## Abstract

The spike protein receptor-binding domain (RBD) of SARS-CoV-2 is required for the infection of human cells. It is the main target that elicits neutralizing antibodies and also a major component of diagnostic kits. The large demand for this protein has led to the use of plants as a production platform. However, it is necessary to determine the *N*-glycan structures of an RBD to investigate its efficacy and functionality as a vaccine candidate or diagnostic reagent. Here, we analyzed the *N*-glycan profile of the RBD produced in rice callus. Of the two potential *N*-glycan acceptor sites, we found that one was not utilized and the other contained a mixture of complex-type *N*-glycans. This differs from the heterogeneous mixture of *N*-glycans found when an RBD is expressed in other hosts, including *Nicotiana benthamiana*. By comparing the glycosylation profiles of different hosts, we can select platforms that produce RBDs with the most beneficial *N*-glycan structures for different applications.

## 1. Introduction

Most COVID-19 vaccines and diagnostic kits for immunological testing target the SARS-CoV-2 spike (S) protein, particularly the receptor-binding domain (RBD), which is the viral component that directly interacts with target cells and stimulates virion internalization [1,2]. Given the limited global capacity of traditional fermenter platforms based on mammalian cell lines, plants are considered a more cost-effective alternative for recombinant S protein/RBD production, especially large-scale manufacture for developing-country applications [3,4]. This has resulted in the development of several plant-based COVID-19 diagnostic reagents [5] and vaccine candidates [6,7,8], some of which have advanced quickly to clinical testing [9,10].

One issue that must be considered when testing alternative production platforms is that host cells differ significantly in their capacity for protein glycosylation. This is important in the context of COVID-19 because the glycan structures on the SARS-CoV-2 S protein interact directly with host molecules, in this case angiotensin-converting enzyme 2 (ACE2) on the surface of respiratory epithelial cells, and therefore influence host–pathogen interactions, viral infectivity, and the immunogenicity of vaccines which are intended to elicit neutralizing antibody responses [11,12]. Mammalian cells are usually preferred for pharmaceutical protein manufacture because they generally produce human-compatible glycan structures [13]. Plant-based expression systems produce unique glycosylation profiles that may influence the efficacy and safety of SARS-CoV-2 vaccines and the accuracy of diagnostic kits [14,15].

We previously evaluated the expression of RBDs in transgenic rice lines and confirmed the accumulation of a functional, correctly folded RBD protein in rice callus and seeds [16]. Recombinant proteins produced in rice often display simpler and more homogeneous glycans than the same proteins produced in animal cells, or even in other plants such as tobacco, which may reflect the adaptation of cereal post-translational modification pathways to produce homogeneous seed storage proteins [17]. We therefore evaluated the *N*-glycan profile of the RBD produced in rice callus and compared it to the spike/RBD produced in other heterologous systems. By comparing the glycosylation patterns across different production platforms, one can select those producing or favoring glycans that are compatible with specific applications, such as the development of vaccines and/or diagnostic reagents [18].

## 2. Results

The RBD sequence (194 residues, N331–V524) was expressed in rice callus with a *C*-terminal His_6_ tag to allow for purification by immobilized metal affinity chromatography (IMAC). We recovered 13 transgenic callus lines, all of which contained the *RBD* gene as determined by polymerase chain reaction (PCR) (Figure 1).

Soluble RBD protein levels were measured by Enzyme-Linked ImmunoSorbent Assay (ELISA) based on human ACE2 to ensure only RBDs with the correct tertiary structure were captured (Figure 2). RBDs were present in nine of the thirteen callus lines (~70%), but reached the limit for detection by Western blot in only six cases (Appendix A). The highest yields were observed in lines 7 and 12, which produced 1.80 and 3.50 µg/g fresh weight (fw) of RBDs, respectively (Figure 2). Callus subcultures over a period of more than 3 months confirmed that the *RBD* gene was stable and there was no significant degradation of the RBD protein.

We generated callus material from lines 7 and 12 to proceed with RBD purification by IMAC. Optimal recovery was achieved by elution with 100 mM imidazole (Appendix A) as confirmed by ELISA (Figure 3).

The IMAC procedure achieved a recovery of only ~8%, possibly because the *C*-terminal His_6_ tag was partially buried within the three-dimensional structure of the protein. Large-scale batch purification with ~230 g callus tissue (Appendix A) ultimately yielded 25 μg RBDs in a volume of 7860 μL (Appendix A). The purified RBD protein was used for glycoproteomic analysis to confirm the protein sequence and the *N*-linked glycosylation profile [19]. The RBD sample was digested sequentially with trypsin and Glu-C, and the mixture of peptides and glycopeptides was analyzed by Liquid Chromatography Electrospray Ionization Tandem Mass Spectrometric (LC-ESI-MS/MS). The protein sequence coverage was ~62%, including the His_6_ tag (Figure 4).

The RBD sequence has two canonical *N*-linked glycan acceptor sites at Asn27 and Asn39 (Figure 4). The *N*-glycan profiles and distributions were determined by analyzing the peptide and glycopeptide mixture using a targeted LC-ESI-MS/MS method [19]. Peptides with *N*-glycan diagnostic fragment ions at *m/z* 204 (*N*-acetylglucosamine) and 366 (Man-GlcNAc) were deemed to be glycopeptides. This allowed us to identify three main glycans attached to Asn39 (Table 1). The structure of these glycans was deduced from their molecular weight and data in the literature [20]. We did not identify any glycopeptides associated with Asn27.

Figure 5 shows the MS/MS spectrum and fragment ions of the doubly charged [M+2H/2]^2+^ ion assigned to peptide V37FNATRFASVY47 and linked to complex *N*-glycan Man_3_XylFucGlcNAc_2_. This is a core Man_3_GlcNAc_2_
*N*-glycan bearing an α(1,3)-fucose residue linked to the proximal GlcNAc as well as a β(1,2)-xylose motif linked to the β-Man. The glycopeptide fragment ion at *m/z* 1623.87 confirmed that the α(1,3)-fucose residue is attached to the reducing end of GlcNAc of the chitobiose motif. Moreover, glycan fragment ions containing a xylose residue were observed at *m/z* 660.24 and 822.29.

## 3. Discussion

Plants are economically advantageous for the production of pharmaceutical proteins because they do not need expensive good manufacturing practice (GMP) facilities for upstream production, which can instead be carried out in much less expensive and more scalable greenhouses [21]. Rice is particularly attractive as a large-scale production platform [22,23,24]. However, the glycan profile of recombinant pharmaceutical proteins in rice has in some cases been shown to differ considerably from the glycosylation in mammalian cells, and even from that in other plants [25].

In one informative example, the HIV-neutralizing monoclonal antibody 2G12 was expressed in rice, and its glycan structure was compared to the same antibody produced in maize, tobacco, and Chinese hamster ovary (CHO) cells, the latter being the gold standard for antibody manufacture [16]. Whereas 10–13% of the maize 2G12 heavy chain was devoid of glycans, with the remainder mainly comprising complex-type glycans (if the product was secreted) or oligomannose glycans (if retained in the endoplasmic reticulum, ER) along with a small proportion of single GlcNAc residues [26], the proportion of aglycosylated heavy chain molecules increased to 50% in rice, along with 33% vacuolar-type complex *N*-glycans, 13% single GlcNAc residues, and 4% oligomannose glycans [17]. In tobacco and *Arabidopsis*, 2G12 was predominantly decorated with complex-type glycans if the product was secreted and oligomannose glycans if it was retained in the ER [27,28]. CHO cells were the only host to produce sialylated glycans.

For the RBD protein, we found that one potential glycan site (Asn27) was not utilized and the other (Asn39) contained a mixture of complex-type glycans as usually found on proteins that are secreted to the apoplast or diverted from the Golgi to storage compartments such as the vacuole. In contrast, oligomannose glycans are predominant in ER-retained proteins because the enzymes that transfer GlcNAc onto trimannosyl Man_3_GlcNAc_2_core *N*-glycans are only found in the Golgi apparatus. We used the rice α-amylase *N*-terminal signal sequence to secrete RBDs, and the lack of oligomannose glycans in our RBD samples indicates that this process is efficient in rice callus.

Previous studies looking at the extent of glycosylation on the RBD or entire spike protein have shown considerable differences between production hosts. A direct comparison of recombinant spike proteins produced in tobacco Bright Yellow-2 (BY-2) cells [29] and human embryonic kidney (HEK 293) cells [30] revealed the presence of 115 different glycans, some of which were unique to *N. benthamiana* and BY-2 cells and others (particularly those containing sialic acid residues) unique to the HEK 293 cells. A more recent comparison of stably transformed BY-2 cells with transient expression in *N. benthamiana* showed that both systems were suitable for the production of ER-retained RBD, albeit with differences in yields and timings [5]. Interestingly, the same study also revealed unexpected differences in the *N*-glycan profiles, with BY-2 cells producing a higher proportion of oligomannose glycans than *N. benthamiana*. Indeed, the presence of complex *N*-glycans in *N. benthamiana* despite the use of a KDEL tag to retrieve RBDs to the ER suggests a degree of leakage into the Golgi apparatus, where complex-type *N*-glycans mature. One potential explanation is that transient expression involves the rapid onset of high-level transgene expression, with the potential to overload the KDEL-mediated retrieval signal. The RBD has been produced not only in wild-type *N. benthamiana* but also in ΔXT/FT lines, resulting in glycan structures lacking the plant core glycoepitopes α(1,3)-fucose and β(1,2)-xylose [12,30,31]. A different version of RBD (R319–F541, longer than the N331–V524 protein produced herein) mainly featured complex-type *N*-linked glycans terminating with GlcNAc residues and small quantities of truncated glycans [31]. In agreement with our results, some sites were almost completely occupied (>99%) whereas others were barely glycosylated (<4%), suggesting that the critical asparagine residue is inaccessible. The R319–F541 variant was found to contain predominantly complex-type *N*-glycans when expressed in *N. benthamiana* (GnGn or GnGnXF, depending on which line was used) [32]. However, co-expression of the enzymes GALT1 and FUT13 allowed for the recovery of RBD variants carrying the Lewis^a^ antigen (β1,3-galactose and α1,4-fucose, Le^a^) [32]. The various glycan structures did not appear to influence binding to ACE2, but the plant-derived RBDs lacking α(1,3)-fucose and β(1,2)-xylose showed higher antibody reactivity than the RBDs produced in HEK 293 cells, and the Le^a^ antigen did not interfere with antibody binding.

## 4. Materials and Methods

### 4.1. Genetic Constructs for Rice Transformation

The RBD sequence (194 residues, N331–V524, YP 009724390.1) was obtained from the Wuhan-Hu-1 SARS-CoV-2 isolate (NCBI reference sequence: NC_045512.2) and was codon-optimized for rice and synthesized by GenScript (Piscataway, NJ, USA). The original SARS-CoV-2 signal peptide was replaced with the rice α-amylase signal peptide to direct the RBD to the secretory pathway. A His_6_ tag was added to the *C*-terminal of the polypeptide by PCR using three pairs of primers (Table 2). The forward primer (5′-AAG GGA TCC ATG GGC AAG CAG ATG-3′) and the final reverse primer (5′-GCG CAA GCT TTC AGT GAT G-3′) included BamHI and HindIII restrictions sites, respectively. The RBD expression vector was controlled by the *ZmUbi1* promoter and first intron [33] and *nos* terminator, and the construct was inserted into the pUC57 backbone to create vector pUbi-RBD (Figure 6). The hygromycin B phosphotransferase gene (*hpt*) was used as a selectable marker [34] also in the pUC19 backbone [35].

### 4.2. Transformation and Regeneration of Transgenic Rice Callus

Mature zygotic rice embryos (*Oryza sativa* cv. Bomba, provided by Illa de Riu, Tarragona, Spain) were transformed by direct DNA transfer followed by callus growth and regeneration under selection, as previously described [36].

### 4.3. DNA Extraction and Analysis

DNA was isolated from callus as previously described [37], with some modifications. Briefly, 300 mg of fresh callus was ground in a porcelain mortar under liquid nitrogen and the powder was transferred to a 2 mL Eppendorf tube. The powder was mixed with 750 μL extraction buffer (100 mM Tris-HCl pH 8, 500 mM NaCl, 50 mM EDTA) and 75 μL 20% (*w*/*v*) SDS by vortexing for 10 min. The mixture was incubated at 65 °C for 1 h before we added one volume of phenol-chloroform-isoamyl alcohol (25:24:1), mixed by vortexing, and separated the layers by centrifugation (13,000× *g*, 5 min, room temperature). The supernatant was mixed with 6 μL RNase (10 mg/mL) and incubated for 30 min at 37 °C. After another round of phenol extraction as above, the supernatant was mixed with an equal volume of isopropanol to precipitate the DNA. After centrifugation (10,000× *g*, 10 min, 4 °C), the DNA pellet was washed with 1 mL 70% ethanol and redissolved in 100 μL Millipore water. The DNA was quantified using a NanoDrop 2000c spectrophotometer (Thermo Fisher Scientific, Waltham, MA, USA).

The coding sequence of the *RBD* gene was amplified by PCR in a total volume of 20 µL, containing 2 µL DreamTaq Green Buffer 10× with 20 mM MgCl_2_, 0.2 µL dNTP (10 µM), 0.2 µL each of the forward and reverse primers (10 µM), 0.125 µL DreamTaq DNA polymerase (5 U/µL), and 1000 ng of callus DNA (all reagents from Thermo Fisher Scientific, Waltham, MA, USA) topped up with Millipore water. The template was denatured at 95 °C for 4 min, followed by 30 cycles (denaturation at 95 °C for 45 s, annealing at 60 °C for 45 s, and extension at 72 °C for 30 s), and a final extension step for 5 min at 72 °C. The *RBD* coding sequence was amplified using the same primers described above. The predicted amplicon size was 680 bp. The *hpt* coding sequence was detected using forward primer hpt-1-F (5′-ACT CAC CGC GAC GTC TGT CG-3′) and reverse primer hpt-1-R (5′-GAT CTC CAA TCT GCG GGA TC-3′). PCR products were resolved by 1% agarose gel electrophoresis in 1× Tris-acetate buffer. The transforming plasmids were used as positive controls and DNA from wild-type (WT) callus was used as the negative control.

### 4.4. Protein Extraction

Fresh callus (150 mg) was ground under liquid nitrogen in three volumes (*v*/*w*) of phosphate-buffered saline (PBS, pH 7.4) for ELISA and Western blot analysis, or in five volumes (*v*/*w*) of PBS with 20 mM imidazole (pH 7.4) for RBD purification. Samples were vortexed and centrifuged twice (10,000× *g*, 10 min, 4 °C). The supernatant was collected and the total protein concentration was determined using Bradford’s reagent (Thermo Fisher Scientific, Waltham, MA, USA).

### 4.5. ELISA

MaxiSorp 96-well clear flat-bottom plates (Thermo Fisher Scientific, Waltham, MA, USA) were coated with 50 ng/well of recombinant human ACE2 (Sino Biological, Chesterbrook, PA, USA) overnight at 4 °C. The plates were washed three times with 200 µL/well of PBST (PBS with 0.1% Tween 20) and three times with 100 µL/well of SuperBlock reagent in PBS (Thermo Fisher Scientific, Waltham, MA, USA), before incubation at room temperature for 2 h with 200 µL/well of blocking solution comprising 1% bovine serum albumin (BSA; Sigma-Aldrich, St. Louis, MO, USA) in PBST. After washing the plates another five times with 200 µL/well of PBST, the supernatant containing the protein fraction was diluted in blocking solution, transferred to the plates, and incubated at room temperature for 90 min. The plates were washed five times with 200 µL/well of PBST before adding 100 µL/well of SARS-CoV-2 RBD polyclonal antibody E-AB-V1006 (Elabscience, Houston, TX, USA) diluted 1:5000 in blocking solution. After incubation for 1 h at room temperature, the plates were washed five times with 200 µL/well of PBST and incubated with a horseradish peroxidase (HRP)-conjugated anti-rabbit IgG secondary antibody (Sigma-Aldrich, St. Louis y Burlington, MA, USA; diluted 1:10,000 in blocking solution) at room temperature for 1 h. Finally, the plates were washed five times with 200 µL/well of PBST and the HRP signal was detected by staining with 100 µL/well of 3,3′,5,5′-tetramethylbenzidine (TMB; Thermo Fisher Scientific, Waltham, MA, USA) in the dark at room temperature for 30–45 min or until color appeared. The reaction was stopped by adding 100 µL/well of 0.16 M sulfuric acid. The signal was quantified by measuring the absorbance at 450 nm in a Model 680 microplate reader (Bio-Rad, Hercules, CA, USA). Recombinant SARS-CoV-2 (2019-nCoV) S1 protein (Sino Biological, Beijing, China) was used as a positive control at four different concentrations (0.8, 0.4, 0.08, and 0.04 ng/µL), and WT rice extract and PBS were used as negative controls. The yield of RBDs in rice callus in µg/g FW was calculated as the product of the OD_450_ value (minus the WT value) multiplied by the rice extract dilution factor. This was used in the standard curve and also multiplied by the volume of buffer used for the initial protein extraction to arrive at the yield of each line.

### 4.6. SDS-PAGE and Western Blot Analysis

Callus was ground in PBS and centrifuged three times (10,000× *g*, 20 min, 4 °C). The supernatant containing the protein fraction (30 μg of total protein) was mixed with 4× loading dye (0.1 M Tris-HCl pH 6.8, 10% SDS, 40% glycerol, 0.06% bromophenol blue) and 5 mM dl-dithiothreitol (DTT; Sigma-Aldrich, St. Louis y Burlington, MA, USA) at 95 °C for 5 min before loading. Samples were separated on a 12% acrylamide resolving gel (10% (*w*/*v*) SDS, 375 mM Tris-HCl pH 8.8, 0.5% ammonium persulfate (APS; Sigma-Aldrich, St. Louis y Burlington, MA, USA), 0.05% *N*,*N*,*N*′,*N*′-tetramethylethylenediamine (TEMED; Sigma-Aldrich, St. Louis y Burlington, MA, USA), and a 4% acrylamide stacking gel (10% (*w*/*v*) SDS, 124 mM Tris-HCl pH 6.8, 0.5% APS, 0.1% TEMED) at 115 V for 60 min.

The separated proteins were transferred to a 0.45 µm nitrocellulose membrane (Bio-Rad, Hercules, CA, USA) using a Trans-Blot semi-dry transfer system (Bio-Rad, Hercules, CA, USA) at 22 V for 45 min. The transfer efficiency was checked by staining the membrane with 0.2% (*w*/*v*) Ponceau S (Biophoretics, Sparks, NV, USA). Membranes were blocked twice at room temperature for 30 min with TBST (0.019 M Tris base, 0.15 M NaCl, 0.1% Tween 20) containing 5% (*w*/*v*) fat-free milk and were probed with the primary anti-RBD antibody (SARS-CoV2/2019 nCov Spike RBD; 40592-t62, Sino Biological, Beijing, China) diluted 1:2000 in TBST containing 2% (*w*/*v*) fat-free milk for 12–16 h at 4 °C with agitation. The membrane was then washed three times with TBST at room temperature for 15 min before incubation with the HRP-conjugated anti-rabbit IgG secondary antibody (Sigma-Aldrich, St. Louis y Burlington, MA, USA; diluted 1:20,000 in TBST + 2% (*w*/*v*) fat free milk) at room temperature for 90 min with agitation. This was followed by 3 × 15 min washes in TBST at room temperature and a fourth wash (10 min in TBS) to remove traces of detergent. Protein bands were visualized by incubating with Immobilon Crescendo Western HRP substrate (Merck-Millipore, Burlington, MA, USA) in the dark for 30 min before exposure on a Bio-Rad ChemiDoc XRS^+^ molecular imager. The AP-conjugated goat anti-rabbit IgG secondary antibody (Sigma-Aldrich, St. Louis y Burlington, MA, USA) was diluted 1:30,000 in TBST + 2% (*w*/*v*) fat-free milk before incubating with the membrane at room temperature for 1 h. After four washes as described above for the HRP antibody, protein bands were visualized using SIGMAFAST BCIP/NBT substrate (Sigma-Aldrich, St. Louis y Burlington, MA, USA) before exposure on the molecular imager. Recombinant SARS-CoV-2 (2019-nCoV) RBD-His protein (Sino Biological, Beijing, China) was used as the positive control.

### 4.7. Gel Staining

After electrophoresis, the gel was stained with 0.1% Coomassie Brilliant Blue R250 in 10% acetic acid/50% methanol for the minimum time necessary to visualize the protein bands. The gel was destained by soaking for at least 2 h in 10% acetic acid/50% methanol with at least two changes. If background staining was still present after 2 h, destaining was continued until the background was clear.

### 4.8. Recombinant Protein Purification

Recombinant RBD was purified from fresh callus using Ni-NTA HisPur resin (Thermo Fisher Scientific, Waltham, MA, USA). Briefly, RBD was purified on a gravity-flow column packed with 40 μL HisPur Ni-NTA resin. The resin was washed with 1 mL cold Milli-Q water then equilibrated with 400 μL of equilibration buffer (50 mM NaH_2_PO_4_, 300 mM NaCl, 10 mM imidazole, pH 7.4) and the equilibration buffer was drained from the column. The filtered callus crude extract was added to the column and the flow-through was collected in a 50 mL Falcon tube. The resin was then washed with 1 mL of washing buffer (50 mM NaH_2_PO_4_, 300 mM NaCl, 25 mM imidazole, pH 7.4) and the wash-through was collected in a fresh tube. RBD-His_6_ was eluted by adding 400 μL of elution buffer (50 mM NaH_2_PO_4_ pH 7.4, 300 mM NaCl, and 50–250 mM imidazole in steps to determine the optimal elution concentration). The matrix was regenerated with 400 μL MES buffer and washed with 1 mL cold Milli-Q water as per the initial step described above, allowing the flow-through from the first purification round to be re-applied three times to maximize binding of the RBD protein to the resin. Elution fractions were concentrated by ultrafiltration using Amicon Ultra spin columns with a 3 kDa molecular weight cut-off (EMD Millipore, Darmstadt, Germany), simultaneously replacing the imidazole with 1× PBS (pH 7.4).

### 4.9. Ammonium Sulfate Precipitation

Sequential ammonium sulfate precipitation (25%, 50%, and 75%) was carried out after purifying the RBD protein to enhance the RBD yield. The tubes were stirred at 4 °C for 1 h and then centrifuged (10,000× *g*, 20 min, 4 °C). The pellet was resuspended in PBS buffer, lyophilized, and stored at −20 °C.

### 4.10. Glycosylation Analysis

#### Endoprotease Digestions

The *N*-glycan profile of the RBD was determined as described in [18] with modifications. Briefly, 50 µg of the recombinant protein (recovered as described above) was purified by 12% (*w*/*v*) SDS-PAGE. After staining with Coomassie Brilliant Blue R250, the band corresponding to the RBD protein was cut into small pieces and washed several times with a 1/1 (*v*/*v*) solution of acetonitrile/100 mM ammonium bicarbonate (pH 8). The protein was then reduced with 100 mM dithiothreitol in ammonium bicarbonate (pH 8) for 45 min at 56 °C, and cysteine residues were alkylated with 55 mM iodoacetamide in 100 mM ammonium bicarbonate (pH 8) for 30 min at room temperature in the dark. The gel pieces were digested with trypsin (Promega, Madison, WI, USA) at a ratio 1:20 in 100 mM ammonium bicarbonate (pH 8) and stored at 4 °C for 45 min prior to overnight incubation at 37 °C. The reaction was stopped by heating to 100 °C for 10 min. The gel pieces were subsequently incubated with Glu-C (Roche, Boulogne-Billancourt, France) at a ratio of 1:50 in 200 µL of 50 mM of NaH_2_PO_4_ (pH 8) at 37 °C overnight. Peptides and glycopeptides were recovered from the gel pieces by sequential washing with 50% acetonitrile (*v*/*v*), 5% formic acid (*v*/*v*), 100 mM NH_4_HCO_3_, 100% acetonitrile (*v*/*v*), and 5% formic acid (*v*/*v*). The five elution fractions were combined and dried in a SpeedVac centrifuge (Thermo Fisher Scientific, Waltham, MA, USA).

### 4.11. LC-ESI-MS/MS Analysis

Peptide samples were resuspended in 3% acetonitrile/0.1% formic acid/96.9% water (*v*/*v*/*v*) for analysis on a Q-Exactive Plus mass spectrometer coupled with an Easy nLC II system (both from Thermo Fisher Scientific, Waltham, MA, USA) equipped with a nanoESI source. Peptides were loaded onto an enrichment column (C18 Pepmap100, 5 mm 300 m, 5 µm particle size, porosity 100 Å; Thermo Fisher Scientific, Waltham, MA, USA) and separated on an analytical column (NTCC-360/100-5-153; Nikkyo Technos, Tokyo, Japan) at a flow rate of 300 µL/min. The mobile phase was 0.1% formic acid in water (buffer A) and acetonitrile/0.1% formic acid in water (80/20) (buffer B). A 45 min elution gradient was applied as the following: 0–19 min, 2–55% B; 19–20 min, 55–100% B; 20–30 min, 100% B; and 30–45 min, 2% B. The temperature of the column was set at 40 °C. The mass spectrometer acquisition parameters were the following: 100 ms maximum injection time, 20 s dynamic exclusion time, AGC target 1 × 10^5^, 2 × 10^4^ intensity threshold, 1.6 kV capillary voltage, 275 °C capillary temperature, and full-scan MS 400–1800 *m/z* with a resolution of 70,000 in MS mode and 17,500 in MS/MS mode. The 10 most intense ions were selected and fragmented by high-energy collisional dissociation with nitrogen as the collision gas (normalized collision energy 27 eV). Raw data were used for subsequent spectral analysis. Protein database searches and PTM identification were carried out using the PEAKS studio v10.5 build 20,191,120 proteomics workbench (Bioinformatics Solutions, Waterloo, ON, Canada) with the following specific parameters: enzyme, trypsin, and GluC; maximum missed cleavages of 3; fixed modification, carbamidomethylation; 314 built-in variable modifications (oxidation (M), deamidation (NQ), pyro-Glu from E and Q cited as examples); monoisotopic mass tolerance for precursor ions, 6 ppm; mass tolerance for fragment ions, 0.02 Da; MS scan mode, FT-ICR/Orbitrap; and MS/MS scan mode, linear ion trap. We only considered high-confidence results with a false discovery rate (FDR) < 1 and a −10 log P ≥ 30. For each glycopeptide, we selected for diagnostic ions at *m/z* 204 (*N*-acetylglucosamine) and 366 (Man-GlcNAc) as well as 512 corresponding to plant Le^a^ epitopes, and then manual analysis and annotation were used to determine the peptide mass and glycan sequence.

## 5. Conclusions

We expressed the RBD of the SARS-CoV-2 spike protein in rice callus with the product directed for secretion to the apoplast. Two *N*-linked glycosylation acceptor sites were identified on the molecule, although one carried no glycans while the other was fully occupied by complex-type glycans. This finding reinforces earlier reports that rice exhibits unique glycan profiles for recombinant proteins, differing even from commonly utilized plant-based production hosts such as *N. benthamiana*. The uniformity we observed in protein glycosylation profiles compared to other plants highlights the advantage of using rice callus in this regard. The prevalence of complex glycans indicates that its RBD is effectively secreted and properly processed within its secretory system. Although the rice callus system achieved low expression levels compared to other platforms, the expression levels were stable and it should be possible to improve them in the future by optimizing the *RBD* transgene and/or expression construct and by testing other tags to improve protein recovery during purification. These results provide opportunities for various plant-based systems to be utilized in the production of RBD-based vaccine candidates and diagnostic reagents, each offering a distinctive glycan profile tailored to specific applications.

## Figures and Tables

**Figure 1 ijms-25-04466-f001:**
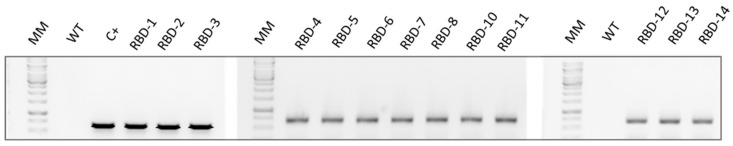
Analysis of genomic DNA from callus lines transformed with the *RBD* gene. MM = 1 kb DNA ladder. WT = negative control. C+ = plasmid DNA (positive control). The anticipated size of the *RBD* amplicon was 680 base pairs (bp).

**Figure 2 ijms-25-04466-f002:**
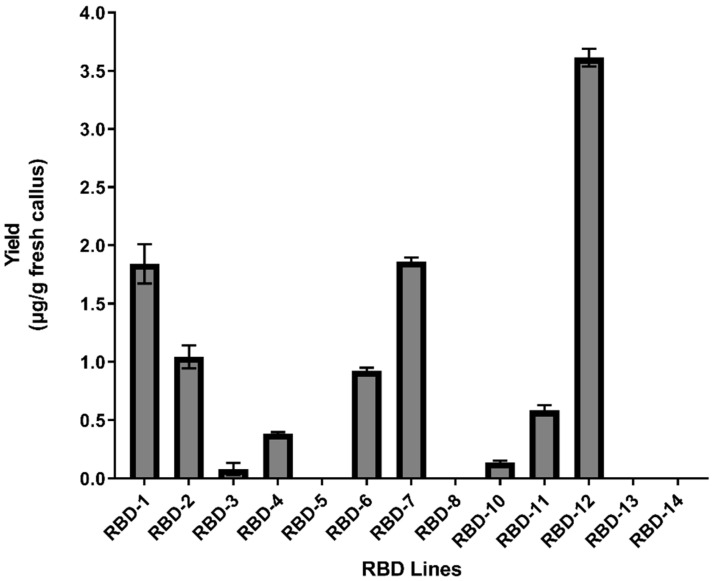
Accumulation of recombinant RBD protein in rice callus lines. The yield of each line is expressed in µg/g fresh callus weight. Data are means ± standard errors (*n* = 3 experiments).

**Figure 3 ijms-25-04466-f003:**
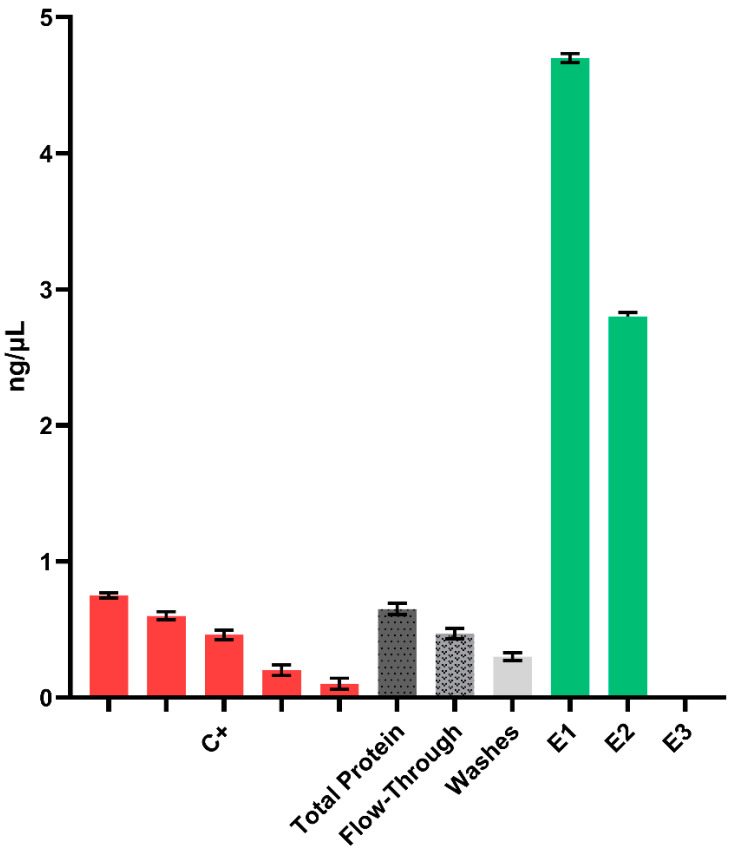
Analysis of fractions collected during RBD purification by ELISA. We analyzed the total protein before purification, as well as the flow-through fractions, washes (all diluted 2/3), and various elution fractions (E1–E3, all diluted by 1/10). Different concentrations of pure recombinant S1 protein produced in baculovirus-infected insect cells were used as a positive control (C+) and to develop the standard curve (red bars).

**Figure 4 ijms-25-04466-f004:**
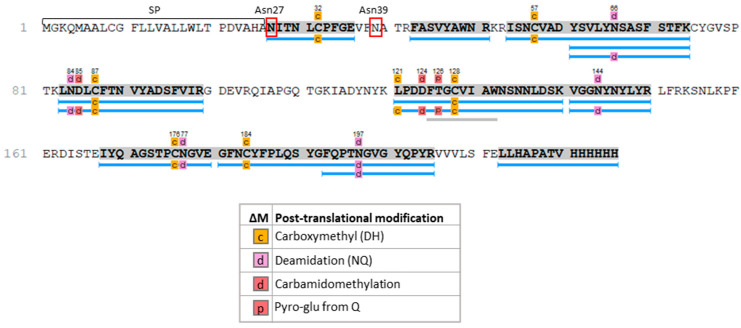
Sequence coverage of the SARS-CoV-2 recombinant RBDs produced in rice callus, as determined by LC-ESI-MS/MS analysis. Gray shading = sequence coverage of the whole protein. Blue bars depict identified peptides. Pale gray bars are sequence tags. SP = rice α-amylase signal peptide. Asn27 and Asn39 represent asparagine residues in consensus *N*-glycosylation sites.

**Figure 5 ijms-25-04466-f005:**
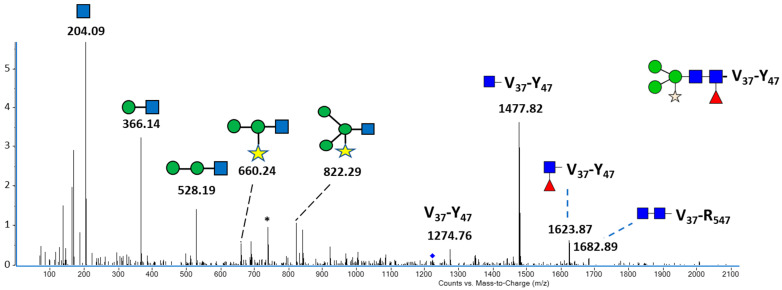
MS/MS spectrum of the [M+2H/2]^2+^ ion at *m/z* 1223.89 assigned to the peptide V_37_FNATRFASVY_47_
*N*-linked to Man_3_XylFucGlcNAc_2_. Main ions were assigned to the glycan and glycopeptide fragments. Symbols: blue square = GlcNAc; green circle = Man; red triangle = Fuc; yellow star = Xyl. * Discharged ion for V_37_-Y_47_ + GlcNAc. The blue diamond indicates the precursor ion at *m/z* 1223.89.

**Figure 6 ijms-25-04466-f006:**
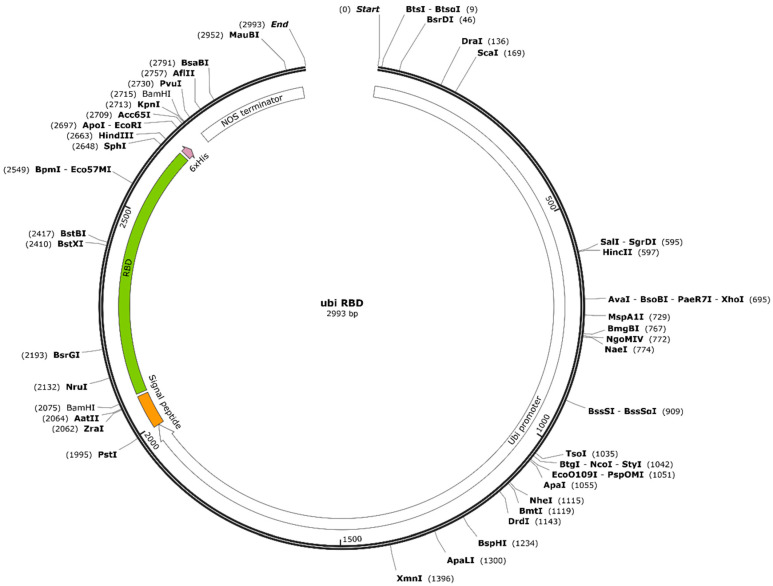
Schematic representation of the RBD plasmid used in transformation. *RBD* (green) was driven by a *ZmUbi1* promoter (white) and has an *Agrobacterium tumefaciens* nopaline synthase (*nos*) terminator (white), rice α-amylase signal peptide (orange), and His_6_ tag (violet).

**Table 1 ijms-25-04466-t001:** Glycopeptides present on the recombinant RBDs expressed in rice and their relative abundance. Blue square = GlcNAc; green circle = Man; red triangle = Fuc; yellow star = Xyl.

Observed Glycopeptide Ions ([M+2H/2]^2+^)	Observed Peptide Ions ([M+H]^+^)	Peptide Sequence	*N*-Glycan Mass	Glycan Structure	%
1142.06	1274.75	VFNATRFASVY	1026.38	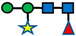	13
1223.09	1274.76	VFNATRFASVY	1188.42	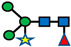	54
1325.13	1274.75	VFNATRFASVY	1391.51	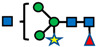	33

**Table 2 ijms-25-04466-t002:** Primers used to amplify and clone the rice *RBD* gene. Restriction sites for BamHI (5′-GGGATCC-3′) and HindIII (5′-AAGCTT-3′) are underlined.

Oligo Name	Sequence (5′ → 3′)
F-RBD-pAL-BamHI (FW 1)	AAGGGATCCATGGGCAAGCAGATG
R-RBD-pAL-HindIII (RV3)	GCGCAAGCTTTCAGTGATG
R-His01 (RV1)	ATGATGATGGCGGGCCCTGCGA
R-His02 (RV2)	TCAGTGATGATGATGATGATGGCGGGCCCT

## Data Availability

Data is contained within the article and Appendix A.

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
