# Peer review of "The SARS-CoV-2 Spike Protein Receptor-Binding Domain Expressed in Rice Callus Features a Homogeneous Mix of Complex-Type Glycans"

_ijms, 2024, doi:10.3390/ijms25084466_

Round 1

Reviewer 1 Report

Comments and Suggestions for Authors

The Authors studied the N-glycan profile of the receptor binding domain (RBD) of the SARS-CoV-2 spike protein. The manuscript is interesting and well-written, however, a few issues should be resolved.

1. Did the Authors investigate the stability of the RBD protein in the rice callus?

2. The map of the vector should be added to the Materials and Methods section.

3. A limitation section should be included in the manuscript.

4. Grammar and punctuation errors should be corrected.

Comments on the Quality of English Language

Grammar and punctuation errors should be corrected.

Author Response

Reviewer 1

(1) Did the authors investigate the stability of the RBD protein in the rice callus?

Authors’ response: We subcultured the callus lines over a period of several months to ensure the transgene was integrated and that transgene expression was stable. During this time, the levels of recombinant protein produced by the callus lines did not change, so we conclude that transgene integration, transgene expression and the RBD protein are stable. We have included a statement to this effect at the beginning of the results section.

(2) The map of the vector should be added to the Materials and Methods section.

Authors’ response: We have added a vector map as requested.

(3) A limitation section should be included in the manuscript.

Authors’ response: We have added a short section describing potential limitations and drawbacks of the experimental approach and the use of rice callus for production in the conclusion paragraph of the manuscript.

(4) Grammar and punctuation errors should be corrected.

Authors’ response: The manuscript has been revised by a professional scientific writer and English native speaker.

Reviewer 2 Report

Comments and Suggestions for Authors

The paper “The SARS-CoV-2 spike protein receptor binding domain expressed in rice callus features a homogeneous mix of complex type glycans” analyzed the N-glycan profile of the RBD produced in rice callus. Here are some shortcomings that need to be further improved or explained.

Comments:

Q1. The repetition rate of the paper is too high, which needs to be carefully modified.

Q2. Did the international community still require and anticipate the development of the SARS-CoV-2 vaccine?

Q3. The method introduction in this paper is too long, while the content and description of the results are obviously insufficient.

Q4. Why is a large amount of data submitted as a supplementary file, resulting in limited analysis of numerous results. How did the authors consider it?

Q5. As it is a supplementary document, some questions are inconvenient to review. As a result, the data provided in the main text is insufficient.

Q6. How to determine the type and linkage of monosaccharides by mass spectrometry?.

Author Response

Reviewer 2

(1) The repetition rate of the paper is too high, which needs to be carefully modified

Authors’ response: This has been addressed in the response to editorial comments, see above.

(2) Did the international community still require and anticipate the development of the SARS-CoV-2 vaccine?

Authors’ response: The work described in the article started at the peak of the COVID-19 pandemic and part of the funding was provided in the context of COVID-19 vaccine research. Although we acknowledge there is now less urgency to develop COVID-19 vaccines, this does not detract from the legitimacy of our work and we are obliged to state its purpose. It should also be acknowledged that new variants of SARS-CoV-2 or other coronaviruses are likely to emerge in the future and our findings remain relevant for this reason.

(3) The method introduction in this paper is too long, while the content and description of the results are obviously insufficient

Authors’ response: The purpose of the methods section is to enable others to repeat the work and we have provided a sufficient (not excessive) amount of detail to ensure this is possible. The purpose of the results section is to describe our experiments and report their outcomes and again we have provided a sufficient amount of detail for this purpose.

 (4) Why is a large amount of data submitted as a supplementary file, resulting in limited analysis of numerous results. How did the authors consider it?

(5) As it is a supplementary document, some questions are inconvenient to review. As a result, the data provided in the main text is insufficient.

Authors’ response: These two comments cover the same topic so we will address them together. The supplementary data are used to present data that are not part of the main story and are not necessary for the reader to understand the work but that may nevertheless be useful to provide additional context. The main story in the paper is the glycosylation of the RBD, and all the data concerning glycans are in the main manuscript. The confirmation of transgene integration and expression are routine aspects of any investigation involving gene transfer to plants and are not needed for readers to understand the glycosylation data. Therefore none of our supplementary data place any limits on our analysis of the key results (the glycans on the RBD). However, we have added the data concerning transgene integration into the main manuscript to accommodate the reviewer’s concerns because these data are at least unique. In contrast, we have left the western blot data in the supplement because these pictures of gels and blots merely support the quantitative data provided by the ELISA experiments and are not needed in the main manuscript. We have also added the table with the primer sequences and the vector diagram into the main manuscript, the latter at the request of the other reviewer.

 (6) How to determine the type and linkage of monosaccharides by mass spectrometry?

Authors’ response: Plant N-linked glycans are composed of a core Man3GlcNAc2 that results from the trimming of mannose residues from oligomannosides synthesized in the ER. This core glycan sequence is common to all N-linked glycans in eukaryotes. In plants, this core is then processed in the Golgi apparatus by the addition of a β(1,2)-xylose on the β-mannose and an α(1,3)-fucose to the proximal GlcNAc. The location of the fucose residue on the first GlcNAc of the chitobiose was confirmed by LC ESI MS/MS (fragment ion of the peptide linked to a Fuc-GlcNAc disaccharide in Figure 5, previously Figure 4). We also added some reporter ions to the same figure for fragments containing the xylose residue and a reference on plant N-linked glycan structures and biosynthesis in the text to support these data.

Round 2

Reviewer 2 Report

Comments and Suggestions for Authors

No additional questions.